# Acetaminophen administration reduces acute kidney injury risk in critically ill patients with Clostridium difficile infection: A cohort study

Yue Liao[1], Yulong Wang[2], Daxue Li[3], Xuewen Qiu[1]*

1 Department of Pharmacy, Chongqing General Hospital, Chongqing University, Chongqing, China,
2 Department of Ophthalmology, Chongqing General Hospital, Chongqing University, Chongqing, China,
3 Department of Breast and Thyroid Surgery, Chongqing Health Center for Women and Children, Women and Children's Hospital of Chongqing Medical University, Chongqing, China

* qiuxuewen@cqu.edu.cn

## Abstract

### Background

Acetaminophen serves as a standard antipyretic and analgesic agent in the intensive care unit (ICU). However, the association between its administration and acute kidney injury (AKI) among critically ill patients remains controversial, particularly lacking research in patients with Clostridioides difficile infection (CDI). Our aim was to explore the potential relationship between early acetaminophen administration and AKI in critically ill patients with concurrent CDI.

### Methods

Using data from the Medical Information Mart for Intensive Care (MIMIC) IV version 2.2 database, we performed a retrospective cohort study. AKI within 7 days of ICU admission was the main outcome that was measured. We utilized multivariable logistic regression models adjusted for potential confounders based on statistical significance and clinical relevance, to investigate the association between acetaminophen exposure and the risk of AKI in patients with CDI. Additionally, subgroup analyses and sensitivity analysis were conducted to assess the robustness of our primary findings.

### Results

The average age of 984 participants was 66.8 ± 16.5 years, and 52.7% (519) were male. The overall proportion of patients who developed AKI was 75.4% (742/984). In patients without and with acetaminophen administration, AKI rates were 79.8% (380/476) and 71.3% (362/508), respectively. Compared to the non-acetaminophen administration group, the risk of AKI was lower in the acetaminophen administration group (absolute risk difference: -8.5%, 95%CI: -13.83% ∼ -3.17%, P < 0.01).After adjusting for potential confounders, acetaminophen administration was associated with a 32% reduction in the risk of AKI (OR = 0.68, 95%CI:0.48 ∼ 0.96, P = 0.027).

**Data Availability Statement:** All relevant data are within the paper and its Supporting Information files.

**Funding:** The author(s) received no specific funding for this work.

**Competing interests:** The authors have declared that no competing interests exist.

## Conclusion

Our study suggests that early acetaminophen administration may offer renal protection by reducing the risk of AKI in critically ill patients with CDI. Prospective, multicenter randomized controlled studies are needed to verify this finding.

## Introduction

Acute kidney injury (AKI) is a common and significant complication encountered in critically ill patients, with a prevalence affecting approximately 30 to 60 percent of individuals in intensive care settings [1–3]. AKI is associated with elevated acute morbidity and mortality, survivors of AKI may also confront hypertension, cardiovascular disease, and stroke, leading to increased post-hospitalization fatality rates as long-term consequences [4]. Risk factors for AKI include sepsis, malaria, hypovolaemia, major surgery, nephrotoxin exposure and opportunistic infections [4, 5].

Clostridioides difficile infection (CDI), with severe clinical presentations, frequently requiring intensive care unit (ICU) admission [6]. CDI Patients exhibit an increased susceptibility to AKI, potentially attributable to diminished renal perfusion, oxidative stress, the effects of Clostridium difficile toxins, and inflammatory processes [7–11]. Toxins produced by Clostridium difficile damage intestinal wall cells, enter the circulation, and trigger a systemic inflammatory response [12, 13]. The subsequent increase in proinflammatory cytokines such as TNF-α and IFN-γ, can further causes multi-organ damage [14], significantly elevating mortality rates and contributing to the deterioration of renal function [7, 15]. Preventing these toxin-mediated inflammatory injuries during CDI is crucial for protecting vital organs, including the kidneys, thereby improving overall disease outcomes [11].

Acetaminophen serves as a standard antipyretic and analgesic agent in ICU [16, 17]. It exerts its therapeutic effects by suppressing the cyclooxygenase activity of COX-1 and COX-2, thereby reducing the synthesis of prostaglandins (PGs). Additionally, acetaminophen inhibits other peroxidases, including myeloperoxidase, resulting in a decrease in the formation of halogenated oxidants, which may contribute to the slowing down of inflammation development [18]. Moreover, studies suggest that acetaminophen may protect kidney function, potentially by mitigating lipid peroxidation induced by acellular hemoglobin [19]. Previous studies have shown that acetaminophen administration was associated with reduced AKI [20–22], while others indicated a scarce association between them [23, 24]. This variability can be attributed to differences in patient demographics, the timing of acetaminophen administration, and the methodological constraints across various research designs. Nonetheless, a consistent theme across these investigations is the ongoing research interest in the potential renal protective effects of acetaminophen. However, limited clinical studies have assessed the impact of acetaminophen exposure on AKI in CDI patients. Against this backdrop, this cohort study aims to explore the relationship between early acetaminophen administration and AKI among CDI patients in the ICU.

## Materials and methods

We enrolled patients with CDI from the MIMIC-IV (Medical Information Mart for Intensive Care IV, version 2.2) database of the Massachusetts Institute of Technology (MIT). MIMIC-IV contains detailed medical data from over 70,000 adult ICU admissions at Beth Israel Deaconess

Medical Center in Boston between 2008 and 2019, including patient measurements, orders, diagnoses, procedures, and treatments [25]. Author Yue Liao secured the requisite permissions to utilize the dataset (certification number 60723603). The study's methodology and reporting adhered to the guidelines set forth by the Strengthening the Reporting of Observational Studies in Epidemiology (STROBE) initiative [26].

## Study population

Patients with CDI met the criteria for our study. The diagnosis of CDI was based on International Classification of Diseases (ICD)-9/10 guidelines, including ICD-9 code 00845 and ICD-10 codes A047, A0471, A0472. To uphold the study's integrity and ensure the robustness of our findings, stringent Exclusion Criteria were implemented: (1) only the first hospital and first ICU admission was considered, data of patients on second or more ICU admissions would be excluded; (2) ICU stays less than 24 hours; (3) the age of participants under 16 years old; (4) missing medical treatment data.

## Definitions

AKI was diagnosed and staged according to the Kidney Disease Improving Global Outcomes (KDIGO) criteria, which provide a standardized, evidence-based approach with superior diagnostic and prognostic accuracy, and broader acceptance in the global medical community compared to other classifications [27]. Consistent with these criteria, AKI was defined as an absolute increase in serum creatinine of $\geq$ 0.3 mg/dL within 48 hours or$\geq$ 1.5 times baseline within 7 days after admission date [28]. Acetaminophen exposure was defined as using at least 1 dose of any form of acetaminophen given within first day after ICU admission.

## Covariates

We extracted all variables from the MIMIC-IV database using Structured Query Language (SQL) with PostgreSQL. Demographic variables, including age, sex, were obtained. Vital signs in this study also included, such as systolic blood pressure (SBP), diastolic blood pressure (DBP), temperature, heart rate, and respiratory rate. Comorbidities, including hypertension, diabetes, myocardial infarct, congestive heart failure, chronic pulmonary disease, and renal disease, were also included for analysis based on the recorded from the database. Laboratory variables, including white blood cell (WBC) count, platelet count, hemoglobin, calcium, potassium, glucose, creatinine, and blood urea nitrogen (BUN). The most severe vital sign readings and laboratory test results from the first day were recorded. Clinical severity on admission was examined using the Sequential Organ Failure Assessment (SOFA) scores. Vasopressors use was collected after ICU admission.

## Outcomes

The primary outcome measured was AKI, with the secondary outcome being severe AKI, both identified within 7 days post-ICU admission, severe AKI was defined as stages 2 or 3 AKI according to the KDIGO definition [28].

## Statistical analysis

We presented normally distributed continuous data as the mean with standard deviation (SD) and skewed continuous data as the median with interquartile range (IQR). Categorical data were expressed in terms of frequency counts and percentages. The comparison of continuous variables between different groups was performed using the independent samples Student's t-

test for normally distributed data or the Mann-Whitney U-test for non-normally distributed data. For categorical variables, we used the chi-square test or Fisher's exact test, depending on the appropriateness for the dataset.

The impact of acetaminophen exposure on AKI was assessed through logistic regression models, specifically analyzing the odds ratio (OR) and its corresponding 95% confidence interval (CI). The models were adjusted for major covariates. The selection of these confounders was based on clinical relevance, and their relationships with the outcomes of interest or a change in effect estimate exceeding 10%. The percentages of covariates with missing data were less than 25% for all analyses. The missing values of the covariates were imputed via multiple imputations [29].

For secondary outcomes, we utilized multivariable logistic regression models, with covariate adjustment conducted in the same manner as in the primary outcome analysis. We stratified the subgroup analyses by some relevant effect covariates, and significant interaction was considered if P <0.05. Additionally, in the sensitivity analysis, we excluded individuals with renal disease and conducted multivariable logistic regression modeling.

All statistical analyses were conducted utilizing the statistical software packages R 3.3.2 (http://www.R-project.org, The R Foundation) and Free Statistics software version 1.9.1(Beijing, China). A two-tailed test was performed, and a P value <0.05 was considered statistically significant.

## Ethics statement

The research, involving human subjects, obtained ethical approval from the Massachusetts Institute of Technology and Beth Israel Deaconess Medical Center. Since the data was sourced from publicly accessible databases, informed consent was waived. This study was conducted in compliance with national legislation and institutional requirements, which exempted it from the need for written informed consent from participants.

## Results

### Participants

From the MIMIC-IV 2.2 database, we included patients who were diagnosed with CDI and were admitted to the ICU for the first time (N = 1165). We excluded participants who were younger than 16 years or had a stay of less than 24 hours (N = 988). After excluding individuals without medical treatment data, the study ultimately resulted in a total of 984 patients for further analysis. Among them, 508 (51.63%) were acetaminophen users. A flowchart depicting the patient selection protocol for our study is shown in Fig 1.

### Baseline characteristics

Baseline characteristics of the study population are listed in Table 1. The age of all participants was 66.8 ± 16.5 years, and 52.7% (519/984) were male. There were several significant differences in baseline health characteristics between the acetaminophen-exposed and non-exposed groups. Specifically, patients in the acetaminophen-exposed group had a higher average body temperature and a greater prevalence of dementia. Additionally, they exhibited lower arterial oxygen saturation (SPO2) levels. These patients were less likely to have comorbidities such as diabetes, renal disease, and chronic pulmonary disease, potentially reflecting differences in their overall health status. Regarding the development of AKI, we found a relatively high overall occurrence rate of 75.4% (742/984). The incidence of AKI for non-acetaminophen

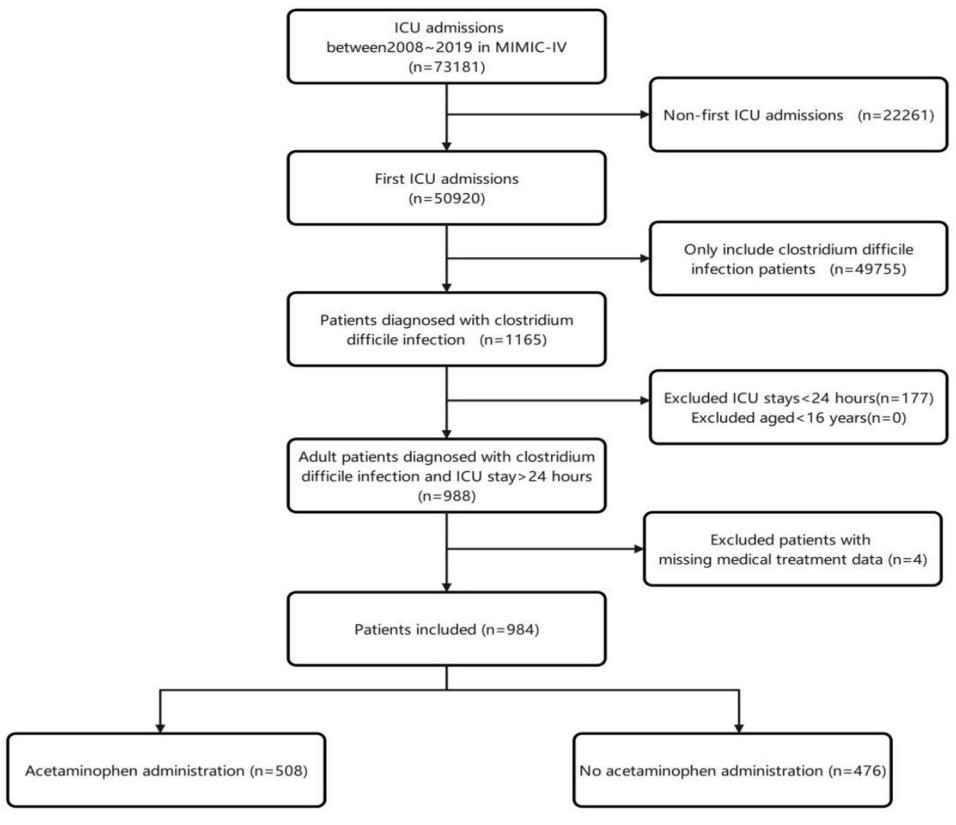

**Fig 1. The flow chart of the study.**

administration group and acetaminophen administration group was 79.8% (380/476) and 71.3% (362/508), respectively.

## Primary outcome

The acetaminophen administration group exhibited a lower risk of AKI compared to the non-acetaminophen administration group, with an absolute risk difference (ARD) of -8.5% (95% CI: -13.83% ~ -3.17%, P < 0.01). This significant reduction in AKI risk associated with acetaminophen administration was observed in both univariable and multivariable logistic regression models (ORs range 0.63 ~ 0.69, p < 0.05 for all). In the fully adjusted model (adjusted covariates of age, sex, heart rate, dbp, sbp, temperature, wbc, platelets, hemoglobin, glucose, bun, creatinine, potassium, calcium, SOFA score, vasopressors use, hypertension, diabetes, myocardial infarct, congestive heart failure, chronic pulmonary disease, and renal disease), the incidence of AKI decreased by 32% (OR = 0.68, 95%CI:0.48 ~ 0.96, P = 0.027) (Table 2).

## Secondary outcomes

After adjusting for potential confounders, participants in the acetaminophen administration group showed a decreased probability of severe AKI (OR = 0.73, 95% CI:0.54 ~ 0.98, P = 0.039) (Table 3).

**Table 1. Baseline characteristics of participants.**

| Covariables | All patients | Acetaminophen used within first day | | P-value |
|---|---|---|---|---|
| | (n = 984) | No (n = 476) | Yes (n = 508) | |
| Age(years) | 66.8 ± 16.5 | 66.9 ± 15.7 | 66.7 ± 17.3 | 0.845 |
| Sex, n (%) | | | | 0.448 |
| Female | 465 (47.3) | 219 (46) | 246 (48.4) | |
| Male | 519 (52.7) | 257 (54) | 262 (51.6) | |
| Body mass index, (kg/m2) | 28.2 ± 9.6 | 27.7 ± 9.7 | 28.7 ± 9.5 | 0.202 |
| Heart rate (bpm) | 91.2 ± 17.2 | 90.4 ± 17.1 | 91.8 ± 17.3 | 0.207 |
| SBP (mmHg) | 115.0 ± 15.6 | 114.6 ± 15.8 | 115.4 ± 15.4 | 0.416 |
| DBP (mmHg) | 61.8 ± 11.2 | 62.2 ± 11.5 | 61.4 ± 10.9 | 0.234 |
| Respiratory rate (breaths/min) | 20.3 ± 4.1 | 20.1 ± 4.1 | 20.4 ± 4.2 | 0.161 |
| Temperature (˚C) | 37.5 ± 0.8 | 37.3 ± 0.3 | 37.7 ± 0.9 | < 0.001 |
| SPO2(%) | 96.9 ± 2.1 | 97.1 ± 2.0 | 96.8 ± 2.3 | 0.029 |
| WBC ($\times 10^9$/L) | 14.8 (9.6, 21.1) | 14.8 (9.1, 20.4) | 14.8 (9.8, 21.5) | 0.257 |
| Platelet ($\times 10^{12}$/L) | 176.0 (115.0, 251.0) | 173.0 (114.5, 241.0) | 181.0 (117.8, 264.0) | 0.149 |
| Hemoglobin (g/L) | 9.7 ± 2.1 | 9.7 ± 2.2 | 9.7 ± 2.0 | 0.739 |
| Glucose (mg/mL) | 140.8 ± 47.0 | 142.4 ± 48.0 | 139.3 ± 46.2 | 0.299 |
| Albumin (g/mL) | 2.9 ± 0.7 | 2.9 ± 0.7 | 2.9 ± 0.7 | 0.137 |
| ALT (IU/L) | 25.0 (15.0, 58.0) | 26.0 (15.0, 62.0) | 24.0 (15.0, 54.0) | 0.135 |
| AST (IU/L) | 42.0 (24.0, 96.0) | 44.0 (25.0, 107.8) | 40.0 (23.0, 83.0) | 0.108 |
| BUN (mg/mL) | 25.0 (16.0, 44.0) | 28.0 (17.0, 49.0) | 23.0 (15.0, 38.0) | < 0.001 |
| Creatinine (mg/mL) | 1.2 (0.8, 2.1) | 1.4 (0.9, 2.5) | 1.1 (0.8, 1.8) | < 0.001 |
| Calcium (mg/mL) | 8.4 ± 0.9 | 8.4 ± 1.0 | 8.4 ± 0.9 | 0.392 |
| Chloride (mmol/L) | 105.8 ± 7.3 | 105.8 ± 7.9 | 105.7 ± 6.6 | 0.735 |
| Potassium (mmol/L) | 3.8 ± 0.6 | 3.9 ± 0.7 | 3.8 ± 0.6 | 0.318 |
| Lactate (mmol/L) | 2.1 (1.3, 3.5) | 2.0 (1.3, 3.5) | 2.2 (1.4, 3.6) | 0.192 |
| PO2 (mmHg) | 79.0 (62.0, 104.0) | 80.0 (63.5, 106.0) | 77.5 (62.0, 103.0) | 0.198 |
| PCO2 (mmHg) | 45.7 ± 13.9 | 45.0 ± 15.0 | 46.5 ± 12.5 | 0.174 |
| SOFA score | 3.0 (2.0, 5.0) | 4.0 (2.0, 5.0) | 3.0 (2.0, 4.0) | 0.002 |
| Charlson comorbidity index | 6.0 (4.0, 8.0) | 7.0 (5.0, 9.0) | 6.0 (4.0, 8.0) | 0.004 |
| Diabetes, n (%) | 298 (30.3) | 160 (33.6) | 138 (27.2) | 0.028 |
| Hypertension, n (%) | 362 (36.8) | 176 (37) | 186 (36.6) | 0.907 |
| Renal disease, n (%) | 272 (27.6) | 150 (31.5) | 122 (24) | 0.009 |
| Chronic pulmonary disease, n (%) | 263 (26.7) | 141 (29.6) | 122 (24) | 0.047 |
| Myocardial Infarct, n (%) | 161 (16.4) | 82 (17.2) | 79 (15.6) | 0.478 |
| Congestive heart failure, n (%) | 321 (32.6) | 159 (33.4) | 162 (31.9) | 0.613 |
| Dementia, n (%) | 42 (4.3) | 12 (2.5) | 30 (5.9) | 0.009 |
| Sepsis, n (%) | 753 (76.5) | 364 (76.5) | 389 (76.6) | 0.969 |
| Vasopressors use, n (%) | 376 (38.2) | 187 (39.3) | 189 (37.2) | 0.502 |
| RRT, n (%) | 79 (54.1) | 53 (60.2) | 26 (44.8) | 0.068 |
| AKI, n (%) | 742 (75.4) | 380 (79.8) | 362 (71.3) | 0.002 |
| AKI stage, n (%) | | | | 0.007 |
| 1 | 119 (12.1) | 56 (11.8) | 63 (12.4) | |
| 2 | 310 (31.5) | 154 (32.4) | 156 (30.7) | |
| 3 | 313 (31.8) | 170 (35.7) | 143 (28.1) | |

Data are presented as the mean ± standard deviation (SD), median (IQR) for skewed variables, and numbers (proportions) for categorical variables.

SBP, systolic blood pressure; DBP, diastolic blood pressure; WBC, white blood count; ALT, alanine aminotransferase; AST, asparate aminotransferase; BUN, blood urea nitrogen; SOFA, Sequential Organ Failure Assessment; RRT, renal replacement treatment.

**Table 2. Multivariable logistic regression to assess the association of acetaminophen administration and acute kidney injury using an extended model approach.**

|         | Odds ratio of acetaminophen used | 95% confidence interval | P-value |
|---------|----------------------------------|-------------------------|---------|
| Model 1 | 0.63 | (0.47 ∼ 0.84) | 0.002 |
| Model 2 | 0.63 | (0.47 ∼ 0.84) | 0.002 |
| Model 3 | 0.66 | (0.49 ∼ 0.9) | 0.008 |
| Model 4 | 0.69 | (0.49 ∼ 0.97) | 0.031 |
| Model 5 | 0.68 | (0.48 ∼ 0.96) | 0.027 |

Adjusted covariates:

Model 1 = acetaminophen administration.

Model 2 = Model 1+ (age + sex).

Model 3 = Model 2 + (hypertension +diabetes +myocardial infarct+ congestive heart failure +chronic pulmonary disease +renal disease).

Model 4 = Model 3+ (wbc +heart rate + dbp+ temperature+

glucose +bun + creatinine + potassium +SOFA score + vasopressors use).

Model 5 = Model 4 + (sbp +platelets +hemoglobin +calcium).

## Subgroup analyses and sensitivity analysis

Subgroup analysis showed that the relationship remained reliable and revealed there is no significant interaction in the different subgroups (all p-values for interaction were >0.05 (Fig 2). We conducted an additional sensitivity analysis, excluding patients with pre-existing kidney disease (n = 272). The results demonstrated that among the remaining 712 patients without kidney disease, the use of acetaminophen was still significantly associated with a reduced risk of AKI (OR = 0.62, 95% CI: 0.42–0.90, P = 0.012) (S1 Table).

## Discussion

In this study, the incidence of AKI in critically ill patients with CDI was 75.4%, which is notably higher than previously reported rates [30]. We investigated the association between early acetaminophen administration and the development of AKI in patients with CDI upon ICU admission, multivariable adjusted models revealed that exposure to acetaminophen was associated with a reduced incidence of AKI. Similar results were observed for the occurrence of severe AKI. Subgroup analyses were performed based on age, sex, comorbidities, and SOFA scores, yielded comparable findings. To further validate our study, we conducted a sensitivity analysis, excluding patients with renal disease, and the results substantiated the robustness of our outcomes.

**Table 3. Multivariable logistic regression to assess the association of acetaminophen administration and severe acute kidney injury using an extended model approach.**

|         | Odds ratio of acetaminophen used | 95% confidence interval | P-value |
|---------|----------------------------------|-------------------------|---------|
| Model 1 | 0.67 | (0.52 ∼ 0.87) | 0.003 |
| Model 2 | 0.67 | (0.52 ∼ 0.87) | 0.003 |
| Model 3 | 0.73 | (0.54 ∼ 0.98) | 0.039 |

Adjusted covariates:

Model 1 = acetaminophen administration.

Model 2 = Model 1+ (age + sex).

Model 3 = Model 2 + (heart rate +dbp +sbp +temperature +wbc +platelets +hemoglobin +glucose +bun +creatinine +potassium +calcium +SOFA score+ vasopressors use +hypertension +diabetes +myocardial infarct +congestive heart failure +chronic pulmonary disease + renal disease)

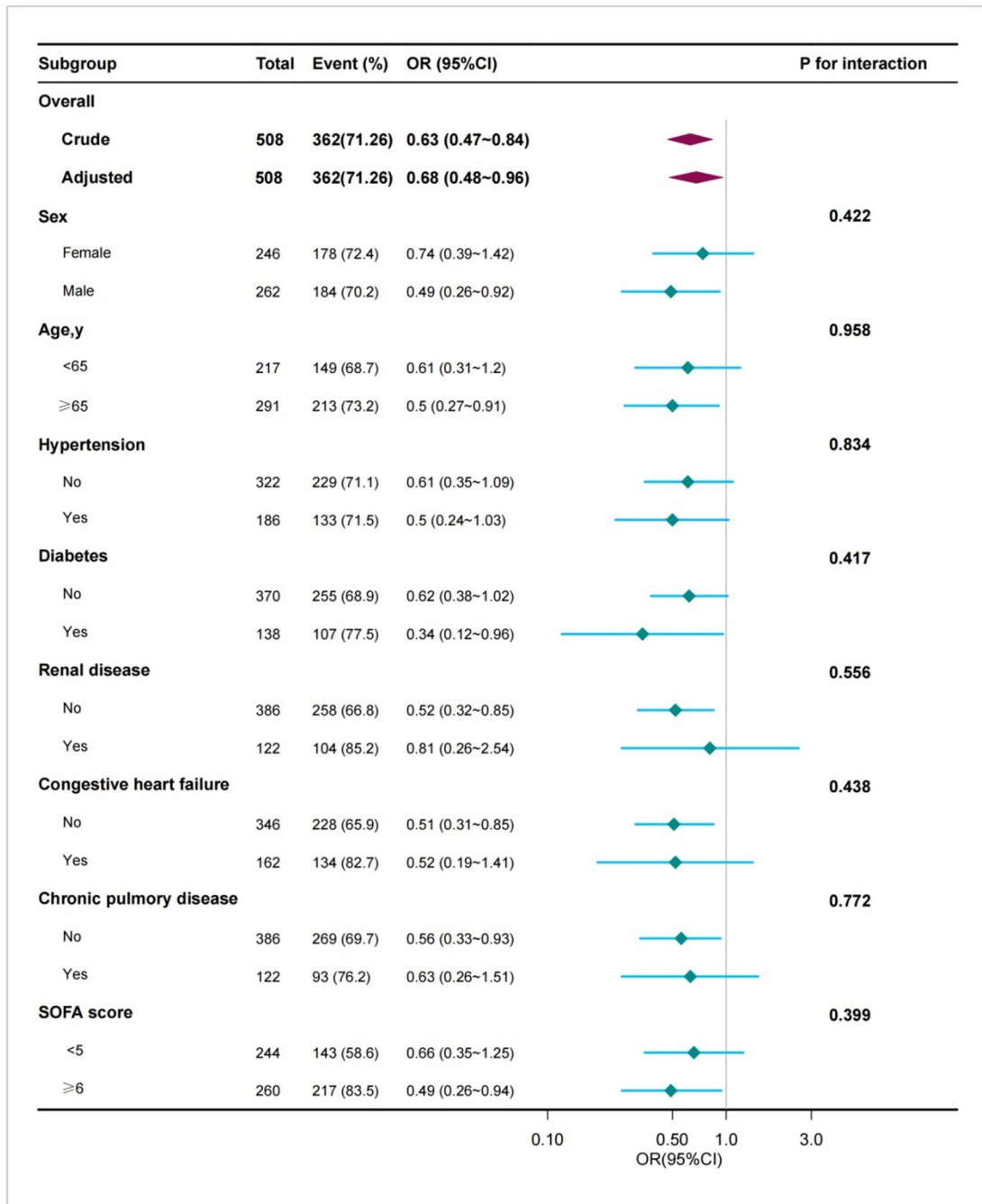

**Fig 2. Forest plot for subgroup analysis for the association between acetaminophen administration and AKI.**

Consistent with our findings, a randomized placebo-controlled trial involving adults with severe sepsis, the administration of acetaminophen within 24 hours of ICU admission reduced oxidative injury caused by elevated plasma cell-free hemoglobin(CFHb) [20], which has been indicated to be correlated with AKI [31]. A recent cohort study investigated the association between acetaminophen administration within 48 hours following cardiac surgery and the incidence of severe AKI within the subsequent 7 days. The findings demonstrated a statistically

significant reduction in AKI among individuals in the acetaminophen-exposed group (HR = 0.86; 95%CI 0.79～0.94, P < 0.001), potentially related to acetaminophen's attenuation of lipid peroxidation [21]. Similar results were observed in a prospective cohort study of 598 children with severe malaria, which indicated that the administration of acetaminophen during hospitalization reduced the acute kidney disease (AKD) following AKI (OR = 0.26; 95% CI 0.15～0.44, P < 0.0001), suggesting that acetaminophen mitigates the risk of ongoing renal function deterioration [22]. Severe acute kidney injury are frequently associated with a fatal prognosis [32, 33]. In our study, after adjusting for confounding variables, the reduced risk of severe acute kidney injury associated with acetaminophen exposure remained statistically significant.

On the other hand, Patanwala et al. conducted a retrospective cohort study suggesting that the use of acetaminophen may not reduce the occurrence or exacerbation of AKI in patients with severe sepsis (OR = 1.2; 95% CI = 0.6～2.4; P = 0.639) [23]. Compared with our study, their sample size was smaller and the study population was different, which may influence the comparability of the results. Furthermore, Patanwala et al. did not further assess the relationship between acetaminophen use and AKI in different subgroups. Hiragi et al. performed a self-controlled case series study indicating that the use of acetaminophen did not lead to the development and progression of AKI. However, their study design had significant differences compared to our cohort study, which may impact the interpretation of the findings. More importantly, they did not further explore whether acetaminophen exposure has a protective effect on renal function, which is a key focus of our research [24]. In a randomized, open-label, controlled trial encompassing 396 individuals diagnosed with malaria, intake of acetaminophen did not demonstrate any renal protective effect across the entire cohort. Notably, different results were observed in a subgroup of patients with severe knowlesi malaria and in those with AKI and hemolysis, patients presenting CFHb level of 77,600 ng/mL or higher, a statistically significant reduction in creatinine levels during the treatment phase was observed in the group exposed to acetaminophen (Coefficient = -0.15; 95%CI = -0.30～0.0065; P = 0.041). This finding seemingly corroborates the hypothesis that acetaminophen may suppress the oxidative renal injury mediated by elevated CFHb levels [19].

In the critically ill patients suffering from CDI, the specific mechanisms through which acetaminophen could mitigate the onset of AKI are yet to be elucidated. Investigations revealed that damage to the intestinal mucosa can precipitate the translocation of gut microbiota and its luminal contents into the systemic circulation, thereby leading to sepsis [12, 34]. During sepsis, increased levels of CFHb may mediate lipid peroxidation, resulting in oxidative damage to renal tubules. Metabolites of this process, such as plasma F2-isoprostanes (F2-IsoPs), can further exacerbate kidney injury by inducing renal vasoconstriction [35]. Acetaminophen, attributed to its ferry1 radical-neutralizing capacity, effectively inhibits hemoglobin-mediated lipid peroxidation. This property is linked to a significant decrease in early plasma F2-IsoPs levels, thereby alleviating the oxidative stress associated with septic conditions [20]. In animal models, Clostridium difficile infection has been shown to induce systemic organ complications in a toxin-dependent fashion, particularly impacting the thymus and renal functions. Renal histopathology in mice infected with toxigenic strains of Clostridium difficile revealed significant macrophage infiltration within the glomeruli (p = 0.0079), a feature notably absent in non-infected controls [11]. Concurrent research has highlighted the capacity of acetaminophen to modulate macrophage cytokine profiles, characterized by a reduction in TNFα secretion and an upregulation of IL-10. These actions may underlie its immunomodulatory role, potentially curtailing the glomerular damage mediated by pro-inflammatory factors [36].

This study has several limitations. First, given its observational and non-randomized nature, we cannot fully exclude the possibility of residual confounding. Despite our comprehensive efforts to adjust for known confounders in our multivariable logistic regression models, the potential for unmeasured variables or selection bias, inherent in a retrospective cohort study, may still influence the results. Second, the single-center nature of our study and the specific demographics of the US population within the MIMIC-IV database may restrict the generalizability of our findings to other settings and populations. We acknowledge the need for further research to validate our results in diverse populations. Third, the absence of detailed data on acetaminophen dosage precludes us from assessing a potential dose-response relationship, which is an area we aim to address in our ongoing research.

## Conclusion

In summary, we provide the first evidence that the administration of acetaminophen within the first day of ICU admission for patients with CDI can reduce the incidence of AKI, and the protective effect of acetaminophen remains significant for the risk of severe AKI. While these findings are promising, future research should include prospective, multicenter randomized controlled trials to definitively establish the causal relationship between early acetaminophen exposure and the prevention of AKI in CDI patients. These trials will be essential in guiding clinical decision-making and treatment guidelines for this patient population.

## Supporting information

**S1 Fig. Collaborative Institutional Training Initiative (CITI program).**
(TIF)

**S1 Table. Multivariable logistic regression was used to assess the association between acetaminophen administration and the risk of AKI excluding patients with renal disease.**
(DOCX)

**S2 Table. Data.**
(CSV)

## Acknowledgments

We thank Dr. Liu Jie (People's Liberation Army of China General Hospital, Beijing, China) for helping with the revision.

## Author Contributions

**Conceptualization:** Yue Liao, Daxue Li.

**Data curation:** Yue Liao.

**Formal analysis:** Yulong Wang.

**Methodology:** Yulong Wang, Xuewen Qiu.

**Project administration:** Xuewen Qiu.

**Software:** Yulong Wang, Daxue Li.

**Writing – original draft:** Yue Liao.

**Writing – review & editing:** Xuewen Qiu.

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
