## [Decision Letter · Decision Letter 0]

7 Oct 2024

PONE-D-24-23392The Relationship Between Acetaminophen Administration and Acute Kidney Injury in Critically Ill Patients with Clostridium difficile Infection: A Cohort StudyPLOS ONE

Dear Dr. Qiu,

Thank you for submitting your manuscript to PLOS ONE. After careful consideration, we feel that it has merit but does not fully meet PLOS ONE’s publication criteria as it currently stands. Therefore, we invite you to submit a revised version of the manuscript that addresses the points raised during the review process.

We look forward to receiving your revised manuscript.

Kind regards,

Diana Laila Ramatillah, PhD

Academic Editor

PLOS ONE

Journal Requirements:

Additional Editor Comments :

This title looks like the Acetaminophen increased Acute Kidney Injury otherwise after we reread the manuscript the result showed the Acetaminophen lower the risk of Acute Kidney Injury.

Reviewers' comments:

Reviewer's Responses to Questions

**Comments to the Author**

1. Is the manuscript technically sound, and do the data support the conclusions?

Reviewer #1: Yes

Reviewer #2: Yes

Reviewer #3: Yes

Reviewer #4: Partly

2. Has the statistical analysis been performed appropriately and rigorously? 

Reviewer #1: Yes

Reviewer #2: Yes

Reviewer #3: Yes

Reviewer #4: Yes

3. Have the authors made all data underlying the findings in their manuscript fully available?

Reviewer #1: Yes

Reviewer #2: Yes

Reviewer #3: Yes

Reviewer #4: Yes

4. Is the manuscript presented in an intelligible fashion and written in standard English?

Reviewer #1: Yes

Reviewer #2: Yes

Reviewer #3: Yes

Reviewer #4: Yes

5. Review Comments to the Author

Reviewer #1: I wish if the authors can briefly state the proposed possible mechanisms of action of acetaminophen in their manuscript, although the full detailed mechanisms are not fully understood according to the FDA and to try to explain the study results in the light of these mechanisms (e.g how the selectivity of acetaminophen towards COX-1, COX-2 and COX-3 variants can justify the observed results)

Reviewer #2: 1). How does the dosage of acetaminophen affect the risk of acute kidney injury (AKI) in critically ill patients with Clostridium difficile infection (CDI) as observed in the study?

2). What confounding factors were identified in the study, and how did adjusting for these factors influence the results regarding the association between acetaminophen dosage and AKI risk?

3). What are the potential clinical implications of the study's findings on acetaminophen administration and AKI risk in patients with CDI, and how might these findings influence future practices in the management of critically ill patients?

Reviewer #3: Dear Editor and Authors,

I would like to start by commending the authors on their important contribution to understanding the relationship between acetaminophen administration and the risk of acute kidney injury (AKI) in critically ill patients with Clostridioides difficile infection (CDI). The use of a large and well-validated dataset like MIMIC-IV adds strength to the findings, and the statistical methods used are robust and appropriate for the study design. The work is both timely and clinically relevant, given the ongoing challenge of AKI management in critically ill populations. I believe this study has the potential to impact clinical practice, and with some revisions, it will be even stronger.

Overall, this is a strong manuscript with valuable clinical insights. I hope the suggestions provided will help enhance its clarity and impact. Thank you again for your hard work and contribution to this important area of research.

Overall, this is a strong manuscript with valuable clinical insights. I hope the suggestions provided will help enhance its clarity and impact. Thank you again for your hard work and contribution to this important area of research.

I have provided detailed feedback on various sections of the manuscript to help enhance its clarity and scientific rigor.

Abstract

The abstract concisely summarizes the study but could benefit from some modifications for clarity:

* Background: The sentence "However, the correlation between its administration and acute kidney injury (AKI) among critically ill patients remains controversial" should be more specific. Adding the phrase “particularly in patients with Clostridioides difficile infection (CDI)” will provide clearer context earlier on.

* Methods: The term "multivariate logistic regression" should be pluralized to "multivariate logistic regression models." Additionally, consider briefly mentioning how confounders were selected or adjusted for, to provide a more complete picture.

* Results: The phrase "we found that acetaminophen administration was associated with 32% lower risk" could be more smoothly expressed as "acetaminophen administration was associated with a 32% reduction in the risk of AKI." Furthermore, specifying "95% CI" instead of "CI" is more standard.

* Conclusion: To emphasize clinical relevance, you could rephrase the conclusion as: "Our study suggests that early acetaminophen administration may offer renal protection by reducing the risk of AKI in critically ill patients with CDI."

Introduction

The introduction is well-written, but there are opportunities to strengthen the framing of the problem and the study's hypothesis:

* Clarity of Concepts: While the background on AKI and CDI is solid, the introduction could benefit from a more explicit explanation of the mechanism by which CDI might increase the risk of AKI. Discussing the toxins produced by Clostridioides difficile and how they exacerbate renal injury would help justify the study's focus.

* Mechanistic Hypothesis: The potential mechanisms by which acetaminophen might protect against AKI (e.g., by reducing oxidative stress or inflammation) could be further elaborated. This would better support the rationale for the study and provide a clearer connection between the drug's properties and the expected outcomes.

* Literature Gap: While you highlight that prior studies have shown inconsistent results regarding acetaminophen and AKI, it would be beneficial to explore why these inconsistencies might exist (e.g., differences in patient populations, timing of administration, or methodological limitations of previous studies). This will strengthen the justification for your study.

Methods

The methods section is detailed, but a few areas could use additional clarification or elaboration:

* Study Design and Data Source: It would be helpful to include more details about why MIMIC-IV was chosen as the data source for this study. Highlighting the database's strengths, such as its large size, diversity of patient populations, and detailed clinical variables, would justify its use.

* Outcome Definitions: The use of KDIGO criteria to define AKI is appropriate, but a brief explanation of why this definition was selected (in comparison to other criteria) would provide additional context for readers who may not be as familiar with this classification.

* Covariates: The covariates are thoroughly described, but it would be helpful to explain the rationale for including specific variables like heart rate and blood pressure as potential confounders. This can clarify how you aimed to control for factors that may influence AKI risk.

* Statistical Analysis: While the use of multiple imputation for handling missing data is appropriate, a brief justification for this method would be beneficial. Additionally, mention if any sensitivity analyses were conducted to assess the robustness of your findings in the face of imputed data.

Results

The results section is generally well-presented but could benefit from additional explanations and some restructuring:

* Clarity in Data Presentation: Although you refer to the tables for detailed results, it would be helpful to provide a more thorough narrative summary of key findings. For instance, instead of just directing readers to "See Table 1," briefly describe the most relevant characteristics, such as significant differences in age or baseline health between groups.

* Subgroup Analysis: The interaction effects from the subgroup analysis (Figure 2) are interesting but would benefit from further discussion. For example, why was there no significant interaction effect in certain subgroups? Offering potential explanations (e.g., sample size limitations, differences in baseline characteristics) would provide a more nuanced interpretation.

* Visuals: The inclusion of a patient flow diagram (Figure 1) is helpful. Just ensure that it is clearly referenced in the text and consider briefly summarizing the patient selection process to help readers follow along more easily.

Discussion

The discussion is thoughtful and connects the findings to existing literature, but it could be expanded in certain areas:

* Comparison with Previous Studies: While you effectively compare your findings with previous studies, it would be useful to delve deeper into why your results may differ from those of Patanwala et al. or Hiragi et al. For instance, were there differences in study design, patient populations, or AKI definitions that could explain the discrepancies? Offering potential reasons for these differences will show a critical engagement with the existing body of research.

* Mechanistic Insights: A more detailed discussion of the potential mechanisms by which acetaminophen might reduce AKI would add depth to the paper. Consider expanding on the role of oxidative stress and lipid peroxidation in AKI and how acetaminophen’s pharmacologic properties may counteract these processes.

* Limitations: You mention some limitations, but it would be beneficial to explicitly address additional potential weaknesses of the study. For example, since this is an observational study, residual confounding may still be present despite adjustments. Similarly, the use of retrospective data from MIMIC-IV could introduce bias related to selection or unmeasured variables. Discussing these limitations will add transparency and strengthen the validity of your findings.

* Clinical Implications: Your findings have clear clinical implications, but they could be emphasized further. For instance, should acetaminophen now be recommended as a preventive measure for AKI in CDI patients? While this study provides evidence in favor of its use, you could mention the need for randomized controlled trials to definitively establish causality.

Additional Comments

* Formatting and Writing Style: Ensure that the formatting is consistent throughout, particularly in the use of citations and the presentation of numerical data. For instance, there are some minor issues like misplaced periods in "consequences1–4." or "infection3,5" that should be corrected for consistency.

* Conclusion: Both the abstract and main text conclusions could be strengthened by suggesting future research directions. For example, while acetaminophen shows promise in reducing AKI risk, a call for randomized controlled trials would make the conclusion more forward-looking.

Reviewer #4: Thank you for the opportunity to review the manuscript titled, "The Relationship Between Acetaminophen Administration and Acute Kidney Injury in Critically Ill Patients with Clostridium difficile Infection: A Cohort Study". The manuscript presents a comprehensive analysis of the relationship between early acetaminophen administration and acute kidney injury (AKI) in critically ill patients with Clostridium difficile infection (CDI) admitted into intensive care unit (ICU). While the study addresses a relevant clinical research problem and has strong potential to improve clinical practice, there are some issues and recommendations the authors needs to address to enhance the robustness and quality of their study.

The following are my comments describing the issues and recommendations:

Comments:

1. Title page

I. The corresponding author should provide an ORCID iD in line with PLOS submission guidelines.

2. Introduction

I. The authors cited previous studies with conflicting results but did not provide a thorough analysis of why these discrepancies exist or how their study aims to resolve them. They should provide additional literature review on the mechanism by which acetaminophene is hypothesized to have a protective effect on the kidneys in CDI patients. This would further strengthen the rationale for this study.

3. Material and methods: The study adheres to general scientific principles, but some areas require more clarity and justification.

I. The definition of acetaminophen exposure as administration within the first day of ICU admission is a somewhat narrow definition, as longer durations of exposure might influence outcomes. The authors should clarify the rationale behind this cutoff and why those who were commenced after 24 hours of ICU admission were excluded

II. The authors did not provide information regarding the use of opiods and nonsteroidal anti-imflammatory, which are associated with an increased risk of AKI in the study population. This should be included as part of limitations of this study if the data cannot be extracted from the MIMIC IV version 2.2 database.

III. They should explain why adverse events of acetaminophen administration such hypersensitivity reactions, acute liver failure, etc were not investigated as one of the secondary outcome measures?

4. Data analysis

The statistical analysis is generally sound and adheres to standard practices for cohort studies.

I. Although the subgroup analyses of the covariates including comorbid conditions were done , the interaction terms are not well-explained, and the justification for specific subgroups analysis is also not clear. The authors need to provide more rationale for the subgroups used in the analysis.

II. Excluding patients with renal diseases would have simplified the analysis by reducing the need to adjust for this significant confounder. The effects of acetaminophen would have been clearer in patient without pre-existing kidney problems.

III. There is a heavy reliance on odds ratios, but a computation of the absolute risk differences would help to contextualize the findings of a decrease in incidence of AKI by 32%.

IV. In addition to the logistic regression analysis, the use of cox regression and hazard ratio analysis would have enhanced the robutness of the study findings, especially by providing insights into the timing of AKI onset relative to acetaminophen administration.

5. Results:

I. The unit of measurement for age in this statement ''The age of all participants was 66.8 ± 16.5" was not stated.

II. Each table should be placed in the manuscript file directly after the paragraph in which it was first cited.

6. Reference

I. The authors did not comply with PLOS recommendation of using Vancouver reference style as outlined by the International Committee of Medical Journal Editors (ICMJE).

7. Writing

I. The manuscript did not include page and line numbers in accordance with PLOS submission guidelines.

II. Some typographical and grammatical errors need to be addressed: For eamples 'our aim is to' was use instead of 'our aim was to" in the abstract, adults was spelt as adluts in the result section, "After adjusted for potential confounders" should written as " After adjusting for confounders" in the abstracts. The appropriate use of indefinite and definite articles through out the manuscript.

III. The were numerous use of single spacing between words and

6. PLOS authors have the option to publish the peer review history of their article (what does this mean?). If published, this will include your full peer review and any attached files.

Reviewer #1: **Yes: **Ahmed Ibrahim Mohamed Ibrahim

Reviewer #2: No

Reviewer #3: No

Reviewer #4: **Yes: **Abbas Lawal Ibrahim

---

## [Author Response · Author response to Decision Letter 0]

4 Nov 2024

Dear Editor Diana Laila Ramatillah, PhD and Reviewers：

Thank you for the opportunity to revise our manuscript entitled "Acetaminophen administration reduces acute kidney injury risk in critically ill patients with Clostridium difficile infection: a cohort study" (ID: PONE-D-24-23392) and for the insightful comments provided by the reviewers. We have carefully considered each comment and have made the following revisions to our manuscript:

*Additional Editor Comments :This title looks like the Acetaminophen increased Acute Kidney Injury otherwise after we reread the manuscript the result showed the Acetaminophen lower the risk of Acute Kidney Injury.

Response:We appreciate your feedback.We have updated the title to better reflect our study's findings:“Acetaminophen administration reduces acute kidney injury risk in critically ill patients with Clostridium difficile infection: a cohort study” (original title:The Relationship Between Acetaminophen Administration and Acute Kidney Injury in Critically Ill Patients with Clostridium difficile Infection: A Cohort Study)，which aligns with the primary outcome and study design as detailed in the manuscript. 

Reviewer #1: I wish if the authors can briefly state the proposed possible mechanisms of action of acetaminophen in their manuscript, although the full detailed mechanisms are not fully understood according to the FDA and to try to explain the study results in the light of these mechanisms (e.g how the selectivity of acetaminophen towards COX-1, COX-2 and COX-3 variants can justify the observed results)

Response：Thank you for your valuable suggestions. We recognize the importance of detailing the mechanisms of action of acetaminophen in the paper to enhance the rationale of the study and the interpretation of the results. In the revised manuscript, we have added the following content to the introduction section(Page 4, Line80-87):

Acetaminophen serves as a standard antipyretic and analgesic agent in ICU [16,17]. It exerts its therapeutic effects by suppressing the cyclooxygenase activity of COX-1 and COX-2, thereby reducing the synthesis of prostaglandins (PGs). Additionally, acetaminophen inhibits other peroxidases, including myeloperoxidase, resulting in a decrease in the formation of halogenated oxidants, which may contribute to the slowing down of inflammation development[18]. Moreover, studies suggest that acetaminophen may protect kidney function, potentially by mitigating lipid peroxidation induced by acellular hemoglobin [19].

In the discussion section, we have added the following content(Page 18, Line344-352):

Investigations revealed that damage to the intestinal mucosa may precipitate the translocation of gut microbiota and its luminal contents into the systemic circulation, thereby leading to sepsis [12,33]. During sepsis, increased levels of CFHb may mediate lipid peroxidation, resulting in oxidative damage to renal tubules. Metabolites of this process, such as plasma F2-isoprostanes (F2-IsoPs), can further exacerbate kidney injury by inducing renal vasoconstriction [34]. Acetaminophen, attributed to its ferryl radical-neutralizing capacity, effectively inhibits hemoglobin-mediated lipid peroxidation. This property is linked to a significant decrease in early plasma F2-IsoPs levels, thereby alleviating the oxidative stress associated with septic conditions [20].

Reviewer #2: 

1). How does the dosage of acetaminophen affect the risk of acute kidney injury (AKI) in critically ill patients with Clostridium difficile infection (CDI) as observed in the study?

Response：Thank you for your valuable feedback. The impact of acetaminophen dosage on the risk of acute kidney injury (AKI) due to Clostridium difficile infection (CDI) in critically ill patients is indeed an important issue. We acknowledged this limitation in the original manuscript's limitations section. We attempted to extract dosage information for acetaminophen, but due to database constraints, we could not obtain specific dosage details for acetaminophen. Consequently, we were unable to assess the potential dose-response relationship. We have included this detail in the limitations of the manuscript, which can be found on(Page19, Line373-376)：

Third, the absence of detailed data on acetaminophen dosage precludes us from assessing a potential dose-response relationship, which is an area we aim to address in our ongoing research. 

2). What confounding factors were identified in the study, and how did adjusting for these factors influence the results regarding the association between acetaminophen dosage and AKI risk?

Response：Thank you for your insightful comments. In the multivariable analysis, confounding factor is an important issue,we performed some different statistical models to verify the results’ stability. Following are the details (1 or 2 or 3)[1,2]：

(1) Factors were chosen when adding it to this model and changed the matched odds ratio by at least 10 percent.

(2) For univariate analysis, we adjusted for variables, of which the p values were less than 0.1.

(3) For multivariable analysis, variables were chosen on the basis of previous findings and clinical constraints.

In our study, the adjustment models were as follows:

TABLE 2| Multivariable logistic regression to assess the association of acetaminophen administration and acute kidney injury using an extended model approach .

 Odds ratio of

acetaminophen used 95% confidence interval P-value

Model 1 0.63 (0.47~0.84) 0.002

Model 2 0.63 (0.47~0.84) 0.002

Model 3 0.66 (0.49~0.9) 0.008

Model 4 0.69 (0.49~0.97) 0.031

Model 5 0.68 (0.48~0.96) 0.027

Model 1 = acetaminophen administration.

Model 2 = Model 1+ (age + sex).

Model 3 = Model 2 + (hypertension+ diabetes+ myocardial infarct+ congestive 

heart failure+ chronic pulmonary disease+ renal disease).

Model 4 = Model 3+ (wbc+ heart rate+ dbp +temperature +glucose +bun+ creatinine+

potassium+ SOFA score+ vasopressors use).

Model 5 = Model 4 + (sbp +platelets +hemoglobin +calcium).

By incrementally adding these variables, we ensured that our findings regarding the association between acetaminophen usage and the prevention of AKI were robust and not attributable to any single confounding factor. 

However, due to limitations in the database, we could not obtain specific dosage details for acetaminophen. We have included this specific information in the limitations of our manuscript, which can be found on (Page 19, Line373-376):

Third, the absence of detailed data on acetaminophen dosage precludes us from assessing a potential dose-response relationship, which is an area we aim to address in our ongoing research. We appreciate the opportunity to clarify our approach to addressing confounding factors and believe that these detailed adjustments strengthen the conclusions drawn from our study.

3). What are the potential clinical implications of the study's findings on acetaminophen administration and AKI risk in patients with CDI, and how might these findings influence future practices in the management of critically ill patients?

Response：Thank you for your interest in our research, your suggestion is very valuable. Our study suggests that acetaminophen administration is associated with a reduced risk of AKI in patients with CDI. This could support the preferential use of acetaminophen over other antipyretics in this patient group. We've added a discussion on the potential clinical implications and future research directions in our conclusion to address these points (Page 20, Line 382-387):

“While these findings are promising, future research should include prospective, multicenter randomized controlled trials to definitively establish the causal relationship between early acetaminophen exposure and the prevention of AKI in CDI patients. These trials will be essential in guiding clinical decision-making and treatment guidelines for this patient population.”

1. VanderWeele TJ. Principles of confounder selection. Eur J Epidemiol. 2019;34: 211–219. doi:10.1007/s10654-019-00494-6

2. Greenland S. Invited commentary: variable selection versus shrinkage in the control of multiple confounders. Am J Epidemiol. 2008;167: 523–529; discussion 530-531. doi:10.1093/aje/kwm355

Reviewer #3: Dear Editor and Authors,

I would like to start by commending the authors on their important contribution to understanding the relationship between acetaminophen administration and the risk of acute kidney injury (AKI) in critically ill patients with Clostridioides difficile infection (CDI). The use of a large and well-validated dataset like MIMIC-IV adds strength to the findings, and the statistical methods used are robust and appropriate for the study design. The work is both timely and clinically relevant, given the ongoing challenge of AKI management in critically ill populations. I believe this study has the potential to impact clinical practice, and with some revisions, it will be even stronger.

Overall, this is a strong manuscript with valuable clinical insights. I hope the suggestions provided will help enhance its clarity and impact. Thank you again for your hard work and contribution to this important area of research.

I have provided detailed feedback on various sections of the manuscript to help enhance its clarity and scientific rigor.

Abstract

The abstract concisely summarizes the study but could benefit from some modifications for clarity:

* Background: The sentence "However, the correlation between its administration and acute kidney injury (AKI) among critically ill patients remains controversial" should be more specific. Adding the phrase “particularly in patients with Clostridioides difficile infection (CDI)” will provide clearer context earlier on.

Response：Thank you for your insightful suggestion, we agree that incorporating more detailed information will enhance the clarity of our narrative. In deference to your guidance, we have made the following modifications to the original text:

Background: However, the association between acetaminophen administration and acute kidney injury (AKI) among critically ill patients remains controversial, particularly lacking research in patients with Clostridioides difficile infection (CDI). 

* Methods: The term "multivariate logistic regression" should be pluralized to "multivariate logistic regression models." Additionally, consider briefly mentioning how confounders were selected or adjusted for, to provide a more complete picture.

Response：Thank you for your constructive suggestions. We have implemented the following updates in our revised manuscript based on your suggestions:

We utilized multivariable logistic regression models adjusted for potential confounders based on statistical significance and clinical relevance, to investigate the association between acetaminophen exposure and the risk of AKI in patients with CDI. Additionally, subgroup analyses and sensitivity analysis were conducted to assess the robustness of our primary findings.

* Results: The phrase "we found that acetaminophen administration was associated with 32% lower risk" could be more smoothly expressed as "acetaminophen administration was associated with a 32% reduction in the risk of AKI." Furthermore, specifying "95% CI" instead of "CI" is more standard.

Response：We sincerely appreciated the valuable feedback. Based on your suggestion, we have revised the sentence for greater clarity and specified "95% CI" instead of "CI".

Results: After adjusting for potential confounders, acetaminophen administration was associated with a 32% reduction in the risk of AKI (OR = 0.68, 95%CI:0.48~0.96, P=0.027).

* Conclusion: To emphasize clinical relevance, you could rephrase the conclusion as: "Our study suggests that early acetaminophen administration may offer renal protection by reducing the risk of AKI in critically ill patients with CDI."

Response：Thank you for your constructive suggestion to enhance the clinical relevance of our conclusions. We have rephrased the conclusion to better emphasize the potential clinical implications of our study as per your recommendation:

Conclusion: Our study suggests that early acetaminophen administration may offer renal protection by reducing the risk of AKI in critically ill patients with CDI. Prospective, multicenter randomized controlled studies are needed to verify this finding.

Introduction

The introduction is well-written, but there are opportunities to strengthen the framing of the problem and the study's hypothesis:

* Clarity of Concepts: While the background on AKI and CDI is solid, the introduction could benefit from a more explicit explanation of the mechanism by which CDI might increase the risk of AKI. Discussing the toxins produced by Clostridioides difficile and how they exacerbate renal injury would help justify the study's focus.

Response：Thank you for your constructive suggestions. We have expanded the introduction to include a detailed discussion on the potential mechanisms by which CDI might increase the risk of AKI. Specifically, we have added a section that describes the toxins produced by Clostridioides difficile, and their potential role in exacerbating renal injury. This new section also discusses how these toxins contribute to inflammation, which could lead to AKI (Page4, Line69-74):

CDI Patients exhibit an increased susceptibility to AKI, potentially attributable to diminished renal perfusion, oxidative stress, the effects of Clostridium difficile toxins, and inflammatory processes[7–11]. Toxins produced by Clostridium difficile damage intestinal wall cells, enter the circulation, and trigger a systemic inflammatory response [12,13]. The elevation of proinflammatory cytokines such as TNF-α and IFN-γ further causes multi-organ damage[14], which confers a significantly elevated mortality and are associated with progressive deterioration of renal function[7,15].

* Mechanistic Hypothesis: The potential mechanisms by which acetaminophen might protect against AKI (e.g., by reducing oxidative stress or inflammation) could be further elaborated. This would better support the rationale for the study and provide a clearer connection between the drug's properties and the expected outcomes.

Response：Thank you for your suggestion. We have expanded our discussion to include an overview of the potential mechanisms by which acetaminophen may protect against AKI (Page4, Line80-87):

Acetaminophen serves as a standard antipyretic and analgesic agent in ICU [16,17]. It exerts its therapeutic effects by suppressing the cyclooxygenase activity of COX-1 and COX-2, thereby reducing the synthesis of prostaglandins (PGs). Additionally, acetaminophen inhibits other peroxidases, including myeloperoxidase, resulting in a decrease in the formation of halogenated oxidants, which may contribute to the slowing down of inflammation development [18]. Moreover, studies suggest that acetaminophen may protect kidney function, potentially by mitigating lipid peroxidation induced by acellular hemoglobin [19].

* Literature Gap: While you highlight that prior studies have shown inconsistent results regarding acetaminophen and AKI, it would be beneficial to explore why these inconsistencies might exist (e.g., differences in patient populations, timing of administration, or methodological limitations of previous studies). This will strengthen the justification for your study.

Response：Thank you for your insightful comment. You are correct that prior studies have presented inconsistent findings regarding the relationship between acetaminophen and AKI. We have added a section in the discussion that explores potential reasons for these inconsistencies, and demonstrates the necessity of our study (Page5, Line90-98):

Previous studies have shown that acetaminophen administration was associated with reduced AKI[20–22], while others indicated a scarce association between them [23,24]. This variability can be attributed to differences in patient demographics, the timing of acetaminophen administration, and the methodological constraints across various research designs. Nonetheless, a consistent theme across these investigations is the ongoing research interest in the potential renal protective effects of aceta

---

## [Editor Report · Decision Letter 1]

19 Nov 2024

Acetaminophen administration reduces acute kidney injury risk in critically ill patients with Clostridium difficile infection: a cohort study

PONE-D-24-23392R1

Dear Dr. Qiu

We’re pleased to inform you that your manuscript has been judged scientifically suitable for publication and will be formally accepted for publication once it meets all outstanding technical requirements.

Kind regards,

Diana Laila Ramatillah, PhD

Academic Editor

PLOS ONE
---

## [Editor Report · Acceptance letter]

12 Dec 2024

PONE-D-24-23392R1 

PLOS ONE

Dear Dr. Qiu, 

I'm pleased to inform you that your manuscript has been deemed suitable for publication in PLOS ONE. Congratulations! Your manuscript is now being handed over to our production team.

Kind regards, 

on behalf of

Prof Diana Laila Ramatillah 

Academic Editor

PLOS ONE